# Harmonizing Magnetic Mitohormetic Regenerative Strategies: Developmental Implications of a Calcium–Mitochondrial Axis Invoked by Magnetic Field Exposure

**DOI:** 10.3390/bioengineering10101176

**Published:** 2023-10-10

**Authors:** Alfredo Franco-Obregón

**Affiliations:** 1Department of Surgery, Yong Loo Lin School of Medicine, National University of Singapore, Singapore 119228, Singapore; afo@nus.edu.sg; Tel.: +65-6777-8427 or +65-6601-6143; 2Institute of Health Technology and Innovation (iHealthtech), National University of Singapore, Singapore 117599, Singapore; 3Biolonic Currents Electromagnetic Pulsing Systems Laboratory (BICEPS), National University of Singapore, Singapore 117599, Singapore; 4NUS Centre for Cancer Research (N2CR), Yong Loo Lin School of Medicine, National University of Singapore, Singapore 117597, Singapore; 5Healthy Longevity Translational Research Programme, Yong Loo Lin School of Medicine, National University of Singapore, Singapore 119228, Singapore; 6Department of Physiology, Yong Loo Lin School of Medicine, National University of Singapore, Singapore 117593, Singapore; 7Nanomedicine Translational Research Programme, Centre for NanoMedicine, Yong Loo Lin School of Medicine, National University of Singapore, Singapore 117544, Singapore

**Keywords:** regenerative medicine, bioengineering, magnetoreception, bioelectromagnetics, magnetic field therapy, tissue engineering

## Abstract

Mitohormesis is a process whereby mitochondrial stress responses, mediated by reactive oxygen species (ROS), act cumulatively to either instill survival adaptations (low ROS levels) or to produce cell damage (high ROS levels). The mitohormetic nature of extremely low-frequency electromagnetic field (ELF-EMF) exposure thus makes it susceptible to extraneous influences that also impinge on mitochondrial ROS production and contribute to the collective response. Consequently, magnetic stimulation paradigms are prone to experimental variability depending on diverse circumstances. The failure, or inability, to control for these factors has contributed to the existing discrepancies between published reports and in the interpretations made from the results generated therein. Confounding environmental factors include ambient magnetic fields, temperature, the mechanical environment, and the conventional use of aminoglycoside antibiotics. Biological factors include cell type and seeding density as well as the developmental, inflammatory, or senescence statuses of cells that depend on the prior handling of the experimental sample. Technological aspects include magnetic field directionality, uniformity, amplitude, and duration of exposure. All these factors will exhibit manifestations at the level of ROS production that will culminate as a unified cellular response in conjunction with magnetic exposure. Fortunately, many of these factors are under the control of the experimenter. This review will focus on delineating areas requiring technical and biological harmonization to assist in the designing of therapeutic strategies with more clearly defined and better predicted outcomes and to improve the mechanistic interpretation of the generated data, rather than on precise applications. This review will also explore the underlying mechanistic similarities between magnetic field exposure and other forms of biophysical stimuli, such as mechanical stimuli, that mutually induce elevations in intracellular calcium and ROS as a prerequisite for biological outcome. These forms of biophysical stimuli commonly invoke the activity of transient receptor potential cation channel classes, such as TRPC1.

## 1. Introduction

Electromagnetism is an authentic developmental force. Life on Earth evolved in an electromagnetic realm. The evolving mechanisms of life adapted themselves to exploit the ambient forces (gravity, sunlight, geomagnetism) and environmental conditions (oxygen and water enrichment) present during their primordial development. It thus stands to reason that magnetism would have developmental consequences, an assumption that is backed by a history of experimental observation and clinical practice [1,2,3,4,5,6]. Evidence that magnetism is an authentic developmental stimulus has come from numerous studies showing that shielding cells from all ambient magnetic fields (geomagnetic and man-made) reverts their existing developmental directives, particularly those programs that are invoked by mitochondrial respiration [7,8,9,10,11,12] and calcium homeostasis [13]. It would therefore be justifiably accurate to consider magnetism as a true biological imperative.

In recent years, the enzymatic and genetic pathways activated by extremely low-frequency electromagnetic field (ELF-EMFs) exposure have come into clearer focus [1,5] and commonly invoke calcium signaling [4,14,15,16,17] and oxidative stress [18,19] cascades. These two response limbs are not mutually exclusive [10,19,20,21,22,23,24] but converge at the level of the reciprocal modulation of mitochondrial respiration by calcium [25] and of extracellular calcium entry by mitochondria-derived reactive oxygen species (ROS) [26,27,28,29]. Generally, reviews focusing on the regenerative aspects of magnetic field paradigms invoke the calcium signaling limb [14], whereas reviews focusing on the detrimental aspects of magnetic field exposure more often discuss the oxidative stress component [18,30]. In reality, however, both calcium and ROS can produce either effect [25]. These two seemingly disparate faces of similar low-energy magnetic exposure paradigms can be reconciled within the context of a mitohormetic response cascade requiring cellular stressors to induce survival adaptations, or demise, depending on absolute level.

Mitochondrial respiratory efficiency and stress resilience are adaptive responses to oxidative stress in the form of ROS that are produced during mitochondrial oxidative phosphorylation. The adaptive nature of mitochondrial respiration is encompassed in a process known as mitohormesis [31]. Mitohormesis entails the process whereby low levels of oxidative stress serve as a mechanism to instill survival adaptations, whereas excessive oxidative stress overrides adaptation and stymies survival. With time, the cell is able to adapt and withstand greater levels of oxidative stress with beneficial outcomes.

Mitohormetic survival adaptations include stimulated mitochondriogenesis, enhanced mitochondrial dynamics, and an increase in mitochondrial antioxidant defenses that in combination lead to a reduction in basal apoptosis [32]. Given that magnetic exposure activates the mitochondrial respiration and consequent ROS production that are required for mitohormetic responses, diametrically opposed cellular responses hence may be observed with differences in absolute levels of magnetic field exposure, amplitude, or exposure duration [33]. In support of hypothetical magnetic mitohormesis, it has been shown that the cellular antioxidative defenses are differentially modulated by changes in ELF-EMF exposure [34]. However, mitochondrial respiration per se is not sufficient to instill a survival advantage as it has to be stimulated against the correct transcriptional background that is commonly calcium-sensitive [10,35,36,37,38]. Seemingly disparate magnetic field exposure paradigms have been shown to activate ROS and calcium signaling, suggesting common underlying mechanisms. For instance, the pivotal studies by Usselman et al. [39,40] align with the results of Yap et al. [10] and Wong et al. [41] as they all invoke cell proliferation in association with ROS formation and calcium entry. The main differences in these studies, however, are the magnitude and frequency of magnetic field exposure, MHz magnetic fields in the µT range [39,40] versus kHz magnetic fields in the mT range [10,41], respectively (also see Section 9).

Stress is a developmental determinant [42] that, by design, is a mitohormetic stimulus. The reason for this lies in the fact that any ensuing cellular response, or adaptation, would ultimately require the provision of mitochondrial energy. The mitohormetic nature of electromagnetic fields has been alluded to in previously published literature with the description of discrete “efficacy windows” of biological responsiveness to ELF-EMFs, whereby low levels of oxidative stress (lower amplitude, frequency, or duration of exposure) produce a given biological response and excessively higher levels of oxidative stress (higher amplitude, frequency, or duration of exposure) produce the opposite or stymied response [5,10,18,19,33,34,43,44,45,46,47].

Mitochondria can be considered the stress sensors of the cell. Mitochondria respond to diverse biophysical stimuli [48], including magnetic fields, with ROS and calcium signaling as common response limbs [19,22,48]. The strategy with reference to choosing the best electromagnetic stimulation protocol for a particular application should then become the following: how to administer the appropriate amount of electromagnetic stress to produce a desired developmental response in the context of existing exogenous biophysical stimuli as well as the existing metabolic or senescence status of the cell system in question? For instance, using the same device it was shown that the administration of low levels of magnetically induced oxidative stress provoked beneficial adaptive developmental responses in healthy muscle cells [10], whereas increasing the levels of magnetically induced oxidative stress in metabolically inflamed breast cancer cells undermined their survival while leaving healthy cells unharmed [49,50]. Both developmental responses were correlated with the expression levels of the calcium-permeable Canonical Transient Receptor Potential Channel Type 1, TRPC1, but differed in the ELF-EMF amplitude, 1.5 mT [10] versus 3 mT [50], respectively, and duration of exposure, 10 min [10] versus 60 min [50], respectively. Therefore, in certain therapeutic scenarios, a trade-off may be exploited between effectively overwhelming inflamed tissues with excessively strong or prolonged magnetic field exposure for targeted killing or the suppression of growth, while not harming collateral healthy tissues endowed with sufficient buffering capacity to accommodate the otherwise potentially damaging oxidative stress.

To reiterate, oxidative stress may not be restrictively damaging, if correctly managed. Low levels of oxidative stress are required to instill survival adaptations, according to mitohormetic principals. Therefore, although to the layperson, the safest approach may seem to avoid all stray magnetic fields because of the oxidative stress they may elicit, neutralizing all cellular oxidative stress would also inadvertently preclude cellular adaptations. This, of course, depends on the source of the ROS and the inflammatory status of the mitochondria. For instance, exercise is a form of systemic muscular/mitochondrial stress that is an accepted method to improve human health and survival. The high intake of antioxidant vitamins, however, has been shown to interfere with the establishment of exercise adaptations [51,52,53,54,55]. Excessively neutralizing the ensuing oxidative stress will effectively negate the establishment of metabolic adaptations in response to exercise. Nonetheless, it would be prudent to avoid overexposure to magnetic fields as the induced oxidative stress may prove to be overwhelming and damaging. Any potential danger is thus a matter of absolute exposure level as well as the existing inflammatory status of the cells or tissues in question at the time of exposure. Ambient magnetic fields would hence need to be taken into consideration as they would contribute to the overall oxidative stress produced by an experimental exposure paradigm.

Biological responses will vary depending on the absolute level of electromagnetic exposure, including those emanating from the laboratory environment (tissue culture incubators and tissue culture hoods, etc.) [7,56] as well as the Earth’s standing geomagnetic field [13]. In one notable example, malignant proliferation was stopped and differentiation reinstated upon magnetic isolation of NT2 human pluripotent embryonal carcinoma cells from the Earth’s geomagnetic field and substituted with precise exposure to ELF-EMFs (2.5 µT at 7 Hz for 5 weeks) in a 10 µT static magnetic field background, equal to the cyclotron frequency of the calcium ion (Ca^2+^-ICR) [13]. Importantly, the developmental effects associated with the applied magnetic field were lost in the presence of the geomagnetic field. Stray magnetic fields can vary widely in frequency and amplitude characteristics. Of relevance, constant (72 h) static magnetic field (140 mT) exposure was shown to stimulate cellular proliferation and extracellular calcium entry, yet attenuate mitochondrial oxidative phosphorylation [57], implicating common enzymatic cascades as those recruited by time-variant magnetic fields, yet exhibiting a different response pattern than those typically observed with time-variant magnetic fields. In this case, mitochondrial respiratory depression may have been a mitohormetic indication of overexposure occurring, following the initial proliferation enhancement. Background ambient electromagnetic exposure is a confounding factor that is simply difficult to control [56] and beyond the technical expertise of most groups studying the developmental effects of bioelectromagnetism. Historical failure to converge on a unifying response to magnetic exposure may have been a consequence of variations in the experimental environment and exposure paradigm, rather than the unique enzymatic cascades evoked by distinct electromagnetic exposure regimens.

Experimental strategies aimed at exploiting the developmental effects of ELF-EMFs are highly varied, undermined by synergistic/antagonistic influences and/or stimuli, and commonly focus on outcome measures, rather than validating underlying mechanisms specific to experimental paradigms. Over the past four decades, a multitude of studies examining the effects of ELF-EMF stimulation have appeared utilizing a broad range of repetition frequencies (Hz to MHz range), amplitudes (µT to 100s mT), and exposure durations (minutes to weeks) [5,16,17,18,19,21,22,23,30,33,44,58,59,60,61] (Figure 1), as well as cellular environments (tissue culture plastic or a variety of scaffolds of diverse biophysical characteristics) [61,62,63,64], which will ultimately influence the cumulative level of oxidative stress experienced by the preparation [44]. Concerning the electromagnetic signal, the most varied parameters appear to be the time of exposure, field directionality, and amplitude of the magnetic signal. Often not taken into consideration are the signal symmetry, whether uniphasic (asymmetrical) or biphasic (symmetrical), and the rise time and shape of the applied signal (sine wave, step, or triangular function), which also have been shown to influence cellular response and require an adequate level of technical expertise to implement [10,49,61,62,65,66,67] (Figure 2). For the sake of brevity, a detailed synopsis of the different signal frequencies, amplitudes, and durations of exposure appearing in studies will not be catalogued here as many recent reviews have covered this topic in sufficient detail [5,16,17,18,19,21,22,23,30,33,44,58,59,61].

The conditions of cell cultivation, culture age, density, and senescence status at the time of magnetic exposure and/or assessment vary widely, or are not reported. The ambient temperature experienced by an experimental sample during exposure, and whether exposure was conducted within the confines of a tissue culture incubator or at ambient room temperature, will also influence biological response to magnetic fields as it will influence cellular enzymatic efficiency and, consequently, mitochondrial respiratory efficiency and mitochondrial response to oxidative stress. Finally, the ubiquitous usage of aminoglycoside antibiotics during cell growth is certainly another confounding agent that is often not taken into consideration or is unavoidable in order to prevent bacterial contamination of the culture preparation. In the following sections, I will discuss the exogenous, intrinsic, and experimental influences that could undermine experimental reproducibility.

## 2. Influence of Cell Status on Experimental Outcome to ELF-EMF Exposure

The effect of ELF-EMF exposure is developmental stage-dependent. In vitro osteogenesis was best achieved when MC3T3-E1 osteoblasts were exposed to ELF-EMFs (7 mT at 15 Hz) prior to osteogenic commitment [10,69]. Similarly, the myogenic potency of ELF-EMF (1.5 mT at 15 Hz) exposure was mitigated if applied after the log phase expansion stage of the myoblasts [10]. These effects align with the recognized role of TRPC1 in assisting early progenitor cell proliferation [10,70] and the recently demonstrated role of TRPC1 in magnetoreception [10,71]. Accordingly, sensitivity to ELF-EMFs was shown to follow the developmental time course of TRPC1 expression that is greatest during early myoblast expansion and wanes as differentiation proceeds [10,41,72]. An analogous developmental efficacy of magnetic field exposure was observed during MSC-derived chondrogenesis [65]. On the other hand, magnetoreception was not correlated with the developmental expression of other TRP channel classes, even following their overexpression [10]. On the other hand, TRPC1 vesicular delivery was shown to be necessary and sufficient to reinstate magnetically induced proliferative capacity and mitochondrial respiration in myoblasts that were genetically engineered to be deficient in said processes following CRISPR/Cas9 TRPC1 knockdown [71].

## 3. Calcium Pathways Implicated in Magnetoreception

The molecular machinery responsible for transducing magnetic stimuli into enzymatic and genetic cascades often invokes calcium signaling and its respective calcium permeation pathway(s) [1,4,5,6,15,16,17,23,45,57,58,73]. This developmental attribute of magnetoreception accords with the accepted cellular physiological and developmental importance of calcium homeostasis [74]. Voltage-sensitive [5,6,45,57,73,75], receptor-mediated [16], and mechanosensitive [1] calcium entry pathways have been implicated in magnetoreceptive responses. As calcium channel crosstalk is known to occur and is mediated by changes in cellular membrane potential, energetics, redox status, or store-operated calcium release [74,76], these calcium pathways need not work in isolation of each other, but likely work in combination upon magnetic-related activation of the receptive channel class(es), particularly within the backdrop of cellular mechanosensitive calcium entry pathways [77,78]. One calcium-permeable cation channel has recently received recognition as part of the magnetoreception apparatus, TRPC1 [4,10,15,16,17,20,23,65,71,79,80]. TRPC1 is one of the most ubiquitously expressed of all TRP channels [81,82] and also has been shown to be modulated by both oxidative [26] and mechanical [26,83] stresses. Reciprocally, magnetic induction of TRPC1 activity produces mitohormetic oxidative responses [10,41,50]. TRPC1 also appears to have thermosensory origins, which may further contribute to the cumulative mitohormetic response with respect to ambient temperature [84]. Magnetoreception, mediated by TRPC1, should thus act synergistically with other forms of biophysical stimuli, such as mechanical input, and have developmental consequences. Given its ubiquitous tissue distribution, broad developmental implications, and integrative nature for diverse forms of biophysical stimuli, this review will largely focus on the role of TRPC1-mediated calcium entry in bioelectromagnetic responses. The reader is referred to existing reviews discussing the roles of voltage-gated, ligand-sensitive, and mechanosensitive calcium entry pathways in magnetoreception [6,15,23,45].

## 4. Contributions from Other Forms of Biophysical Stimuli

Diverse biophysical stimuli have been shown to produce developmental mitohormetic responses involving ROS and calcium signaling [48,85]. Provocatively, the switching between osteogenesis and adipogenesis differentiation states was shown to be modulated by the culmination of distinct biophysical stimuli impinging upon bone marrow-derived mesenchymal stem cells with a noted synergism between mechanical and electromagnetic stimuli [86]. This result is consistent with the fact that both mechanical and electromagnetic cellular responses have in common the involvement of TRPC channels upstream of store-operated calcium entry (SOCE)-regulated mitogen-activated protein kinase kinase signaling pathways [10,87,88]. Notably, low levels of mechanical stimulation were shown to favor adipogenesis, whereas higher levels of mechanical stress or magnetic field exposure enhanced osteogenesis, indicating that tissue-specific gene expression is differentially regulated by the degree of cellular stress. The mitohormetic nature of mechanical stimulation is well accepted, invoking both ROS and calcium signaling [89,90,91,92]. In accordance, it was previously shown that the mechanical background provided by distinct scaffolds contributed to the measured ELF-EMF responses [63] in an electromagnetic chondrogenic paradigm dependent on TRPC1 expression levels [65]. The mechanical environment would hence need to be taken into consideration when estimating the mitohormetic outcome of a chosen magnetic exposure strategy.

TRPC channels act as integrators of diverse forms of biophysical stimuli and, in this capacity, function as the sensory receptors of the cell [81]. Nonetheless, whether TRP channels directly sense biophysical stimuli (temperature, pressure, stretch, pH, light, magnetism, etc.) or must interact with subcellular components in order to transduce their respective form of biophysical stimulation into cellular responses remains largely unresolved [93]. Notably, available evidence indicates that TRPC1 serves as a regulator of the other TRPC family members via a process of heteromultimerization [82,94], theoretically uniting their distinct activation modes into a single-channel complex. Accordingly, mechanotransduction [81,95,96], magnetotransduction [10,65,71,80], phototransduction [97,98,99], thermosensation [84], mitogen activation [82,94], phospholipid activation [81,100], SOCE/calcium-sensing [88], and redox-sensing [26,27,28,101] are distinct and synergistic activation modes of TRPC1, or associated “C” family members. The integrative nature of TRPC1 will have profound implications with reference to magnetoreception in the context of other forms of biophysical stimuli (also see Section 9).

## 5. Implications of Aminoglycoside Antibiotic Use

Most electromagnetic studies have been carried out on in vitro systems. The aminoglycoside antibiotics, particularly streptomycin, are commonly employed in such tissue engineering and regenerative medicine paradigms to curtail bacterial contamination. Although their mechanism of action is commonly attributed to the inhibition of bacterial protein translation, collateral toxicity at the level of the “somatic” and mitochondrial ribosomes is also known [102], which will have downstream effects on mitohormetic responses. The aminoglycoside antibiotic antagonism of certain cation channel classes (attenuating channel-mediated calcium entry) represents another form of collateral cytotoxicity that will implicate magnetoreception. Certain TRP channel classes are known to be antagonized by aminoglycosides antibiotics including the TRPM, TRPC, and TRPV subclasses. This will have critical implications as TRP channels can be envisioned as integrators of diverse biophysical stimuli [103,104] of profound cellular developmental importance [105,106]. For instance, calcium entry via the temperature-sensitive TRPM8 channel has been shown to be blocked at doses of streptomycin conventionally used in tissue culture paradigms that, moreover, prevented mitochondrial uncoupled respiration by white adipose tissue [107]. Cell migration is also inhibited by streptomycin-blockade of TRPM7-mediated calcium entry [108]. Streptomycin also acutely blocks TRPV1-mediated thermoreception in neurons [109]. Most notably in the context of magnetoreception, TRPC1 has been shown to be antagonized by the aminoglycoside antibiotics in neurons [110], heart [111,112], and skeletal muscle as well [26,112,113,114]. A somewhat older body of research has shown that the aminoglycoside antibiotics block voltage-gated calcium entry by interacting with the pores of L-type calcium channels [115,116]. For both L-type voltage-gated calcium [115] and TRPV1 [109] channels the potency of aminoglycoside antibiotic channel blockade was roughly neomycin > streptomycin > gentamicin. The functional antagonism of calcium permeation via these cation channel classes by the aminoglycoside antibiotics is immediate and not dependent on the disruption of protein synthesis which would be much more protracted in time course. When considered in the context of the development responses assisted by TRPC1 and L-type voltage-gated calcium channel classes (see Section 3), the presence of streptomycin (blocking calcium permeation via these channels) during magnetic stimulation will alter magnetically promoted developmental responses and lead to an incorrect interpretation of the results.

## 6. Implications of Magnetic Mitohormesis: A ROS-Calcium Mitochondrial Axis

The mitohormetic aspects of magnetic field exposure have been received with alarm by some [30,45,117] but also as an opportunity by others [33,44]. ROS represent a form of adaptive stress [31] that can account for both positive and negative reports depending on the exposure paradigm [23]. Survival-based mitohormetic adaptations improve resistance to oxidative stress, mitochondrial efficiency, and cell survival in association with the activation of the redox-sensitive transcription factor, nuclear factor erythroid 2–related factor 2 (Nrf2), and are particularly well described in muscle adaptations to exercise [32,92,118,119,120,121]. Notably, the administration of ozone (O_3_) to C2C12 mouse myoblasts produced both beneficial (increased Nrf2 and improved cell survival) and detrimental (reduced Nrf2 and diminished cell survival) mitohormetic responses, depending on dose [122]. This effect showed key genetic and developmental parallels to that produced by graded ELF-EMF exposure (Efficacy Window) in the same myoblast cell line [10], providing further evidence of the fundamental contribution that oxidative stress plays in the cellular adaptations to magnetic field exposure. A TRPC1–mitochondrial axis was previously described that was responsible for mediating magnetic mitohormetic responses in myoblasts [10,71], dental pulp stem cells [80], and breast cancer cells [50]. Improved mitochondrial calcium handling represents another mitohormetic adaptation limb [123] that can be linked back to modulated TRPC1 expression [124]. A ROS-calcium mitochondrial axis is thus implicated in establishing magnetic mitohormetic adaptations. Magnetic mitohormesis may thus instead be received as an opportunity, rather than a threat, as the combined effects of oxidative and calcium stresses are context specific and, to a certain extent, under the experimenter’s control, if taken into consideration.

## 7. Paracrine/Secretome Ramifications of Magnetic Stimulation

Secretome release is a common response of the magnetoreception cascade [14,37,41,61,125,126,127]. The secretome response is activated by mitochondrial oxidative stress as a limb of the systemic mitohormetic adaptive response cascade [128,129,130,131]. Given that secretome mobilization is a mitohormetic response, confusion may have arisen from the fact that inflammatory cytokine release is common to both survival-promoting (adaptive) as well as overwhelming (damaging) levels of oxidative stress [128,131], downstream of Nrf2 [131,132] and PGC-1α transcriptional pathways [128,131,133,134,135]. A notable example is the elevated serum levels of IL-6 following exercise [136], magnetic field exposure [10], and disease-associated systemic inflammation [136]. Based on the erroneous assumption that inflammatory cytokines are solely a sign of systemic metabolic dysfunction, an increase in the serum levels of inflammatory cytokines following magnetic exposure may be wrongly interpreted as an indication of systemic inflammation rather than an adaptive response leading to systemic metabolic improvement. In accordance with a mitochondrially induced secretome response limb to magnetic stimulation, Nrf2 and PGC-1α activation have been reported in response to magnetic exposure [10,137]. The secretome response is also facilitated by extracellular calcium entry. Vesicle release is stimulated by calcium entry [138,139], including that mediated by ELF-EMF exposure [140], aligning with the calcium–mitochondrial axis previously described. Given the oxidative nature of magnetic field exposure, however, it may prove to be a more valuable therapeutic strategy to avoid the direct exposure of an inflamed tissue, to avert excessive oxidative stress at the site of injury, and instead to rely on the delivery of regenerative secretome factors from exposed healthy tissues in the vicinity of the injury, which, in essence, invokes a cross-feeding strategy. Finally, the washing away or replacement of the magnetically induced secretome following exposure will preclude differential responses [41], cautioning against the use of microfluidic strategies when attempting to monitor magnetically induced developmental responses.

## 8. Contribution from Background Ambient Magnetic Fields

The existence of background or ambient magnetic fields (static or time-variant) may confound the interpretation of biological responses to an experimentally applied magnetic field. It was recently shown that static (20 mT) and sinusoidal (20 mT at 50 Hz) magnetic fields act cumulatively to induce ROS production and antioxidant (Nrf2) responses, but that static fields are at stimulating proliferation [141]. These disparate responses are reconciled within the context of the same study with their demonstration that the static magnetic fields produced the least amount of oxidative stress, followed by the time-variant fields, and lastly by their combination, which generated the highest levels of ROS. Such an oxidative scenario possesses mitohormetic potential, whereby low levels of ROS are regenerative and higher levels of ROS are developmentally prohibitive. Stray and contaminating magnetic fields would also disrupt the uniformity of an applied field, which may alter developmental efficacy [10,49,61,65,66,67]. Therefore, the existence of uncontrolled differences in distinct experimental environments may potentially produce distinct biological responses from technically identical exposure systems. As overexposure or non-specific exposure to magnetic fields may be inevitable, the contribution of ambient magnetic fields needs to start being taken into more serious consideration [7,39,56,142,143].

## 9. Proposed Mechanisms for Magnetoreception

Two models for magnetoreception have largely withstood the tests of time and are now commonly accepted as authentic behavioral responses, despite minor mechanistic uncertainties. They are the magnetite and the radical pair mechanisms [1,3,144,145,146,147,148,149]. The former predicts that crystals of magnetite composed of iron oxide (Fe_3_O_4_) enable the conversion of magnetic force into cellular responses. Chains of magnetite crystals are often conceptualized in an arrangement with mechanosensitive cation channels. Under the force of an applied magnetic field, the alignment of the magnetite filament is envisioned to change, exerting torsion on membrane-embedded mechanosensitive channels and influencing their gating mechanisms. Such an arrangement would effectively transduce changes in magnetic fields into cation fluxes of enzymatic consequence. In avian behavioral experiments, magnetite-based responses are disrupted by relatively strong magnetic fields (100 mT to 5 T), bringing into question physiological relevance. Nonetheless, single-domain superparamagnetic magnetite particles are theoretically possible and would be capable of aligning under the influence of magnetic fields on the order of the Earth’s geomagnetic field. Magnetite-based molecular complexes (magnetosomes) have been most convincingly shown to play a role in bacterial magnetotaxis [149] as well as in avian magnetic field intensity sensing [147]. Notably, dead bacteria align to an applied magnetic field equally well as living bacteria, suggesting that magnetite also underlies passive (non-enzymatic) responses to magnetic force.

When specifically examined, the radical pair mechanism (RPM) of magnetoreception has been implicated in many ELF-EMF exposure paradigms with varying degrees of contribution, indicating broader magnetic field intensity and frequency response range [150] than previously hypothesized for the RPM with respect to avian navigational magnetoreception [146]. At the heart of the RPM lies a flavoenzyme containing a delocalized electron that is first photoactivated by light. Photoactivation facilitates the reduction of the flavin by a neighboring protein moiety, commonly a cryptochrome, to form a coherent radical pair between the two molecules. The reduced flavoenzyme may also undertake an electron transfer to molecular oxygen (O_2_) to produce superoxide (O_2_^•−^), thereby influencing cellular oxidative status with the formation of another radical pair with molecular oxygen [40]. More recently, evidence has emerged indicating that the paramagnetic Fe-S clusters of Complex 1 of the mitochondrial electron transport chain are directly responsive to magnetism via an RPM mechanism [151]. In response to appropriately matched low-amplitude static and radio-frequency oscillating magnetic fields, cellular proliferation and the production of ROS were enhanced in cancer cells. The RPM thus has several avenues of operation and application to exploit mitohormetic magnetoreception.

Critically, the interconversion of the radical pairs between the singlet (antiparallel spins) or triplet (parallel spins) states is regulated by an external static magnetic field that, in turn, determines the likelihood of the reaction moving in the reverse (returning to reactants) or forward (producing products) directions, respectively. The interconversion of the radical pairs with the flavoenzyme are hence magnetically tunable. Of importance for RPM-mediated magnetoreception is the fact that the orientation of a time-variant magnetic field relative to the background static field differentially modulates radical pair spin interconversion, providing feedback about the orientation of the static magnetic field [40,142].

Most published studies examining the biochemical and cellular responses of the RPM of magnetoreception have done so without the consideration of photons, implying photoactivation by ambient light or, hypothetically, by biophotons [152,153]. It has been shown that following initial photoactivation and the reduction of a flavin moiety to form a radical pair with an electron donor, the flavin’s return to ground state, that is, the loss of an electron from the partially reduced semiquinone to reform the fully oxidized flavin, is associated with a change in blue fluorescence emission when in the presence of an applied magnetic field [154], indicating that magnetic fields may also provide a source of biophotons. This response is governed by magnetically tuned electron spin–selective recombinations as inherent for the RPM. Notably, exposure to violet-blue light has been shown to increase both intracellular calcium and ROS levels [155]. How endogenous biophotons may be generated or how they could influence magnetoreception via the RPM remains to be investigated and understood.

Intriguingly, the magnetite and RPM models may not be mutually exclusive magnetic response pathways as many organisms have been shown to possess both systems [156]. Magnetite biomineralization has been recently shown to exist across phyla [157], and a magnetite molecular complex with cryptochromes (MagR) has been identified that aligns to the geomagnetic field and is conserved across phyla [158]. Functionally, a magnetic orientation system has been described in termites that employ both magnetite and RPM systems to confer light-insensitive and light-sensitive navigation, respectively [159]. Specifically, Gao et al. [159] demonstrated the existence of a comprehensive geomagnetic navigation system requiring the involvement of a magnetite receptor complex, cryptochrome 2 (Cry2), and an odorant-responsive calcium channel, named olfactory co-receptor (Orco) [160,161,162]. In muscle, Cry2 has been implicated in tissue proliferation and differentiation [163] in association with mitochondrial regulatory pathways [164], responses shared with the TRPC1-sensitive, ELF-EMF exposure of muscle [10,41,71]. Magnetite biomineralization has also been identified in muscle [157]. Therefore, key components of the termite magnetic orientation system are expressed in muscle. Moreover, TRPC1 is implicated in store-operated calcium entry (SOCE) [88], and evidence of a SOCE- and flavin-dependent, redox-sensitive, photomechanical transduction process requiring the participation of Cry2 during muscle development has recently emerged [165]. Finally, the genetic knockdown of the entire TRPC family was shown to alter cryptochrome-mediated circadian (“crycadian”) rhythm, cytokine, and metabolic functions [166], congruent with the cellular functions that have been correlated with TRPC1 expression. Cryptochromes are decedents of light-activated DNA-repair enzymes (photolyases), are commonly associated with flavoprotein moieties, and in diverse organisms are known to set circadian rhythms in response to blue light stimulation [84,167]. In some navigational scenarios, therefore, light and magnetic responses appear to converge on a cryptochrome–magnetite–calcium entry system. Although its relationship to the magnetic mitohormetic developmental responses focused on in this report remains to be elucidated, common molecular participants are recruited in each scenario. A generalized magnetoreceptive complex may hence be envisioned wherein a flavoenzyme-activated cryptochrome interacts with a TRP channel member to initiate the signal transduction cascade associated with magnetic stimulation (Figure 3). Although it is provocative to speculate such a scenario, some aspects of the presented model still face theoretical challenges and remain to be experimentally demonstrated.

Evidence in support of TRPC1–cryptochrome functional interactions arises from the fact that the TRPC subfamily holds the highest degree of homology to the original *Drosophila* phototransductive TRP channel [84], the founding member of the entire superfamily [81,93,103]. A regenerative photomodulation protocol applied to dystrophic muscle was shown to stimulate PGC-1α activity, reduce oxidative stress, and attenuate TRPC channel expression [168], showing striking parallels to the ELF-EMF exposure of skeletal muscle cells [10]. TRPC1 has been directly implicated in phototransduction in diverse species and tissues [97,98,99]. Moreover, both magnetoreception [10] and photomechanical transduction [165] can be antagonized by the TRPC channel blocker, 2-APB, or other forms of inhibiting extracellular calcium entry, drawing provocative parallels. TRPC1 may hence comprise part of a magnetic signaling complex, analogous to Orco (olfactory sensory receptor channel) in termites [159], which in collaboration with Cry2 and potentially other molecular partners confers field directionally specific developmental responses.

Evidence also supports that TRPC1 is activated by light as well as by ELF-EMF exposure. Cryptochromes are typically activated by blue light [144,145,146,147]. Indeed, light at 660 nm (red) and 830 nm (near-infrared) and ELF-EMFs (1.6 mT @ 20 Hz) have been shown to synergize to promote wound healing [169]. TRPC1-mediated photomodulatory responses have been reported in response to both blue (420 nm) [170] and red (635 nm) [171] diode laser irradiation. Although TRPC1 vesicular reintroduction appears to be sufficient to reinstate lost magnetoreception in genetically engineered TRPC1-knockdown muscle cells [71], its interaction with other molecular players involved in magnetoreception cannot be discounted. A critical issue is that EMF studies are often carried out under ambient lighting conditions, whereas photomodulatory studies are commonly carried out under conditions of ambient magnetic exposure. The interplay between light and ELF-EMF exposure is an important issue that remains to be resolved. Another open issue is whether the aminoglycoside antibiotics would block photomodulatory responses, particularly if invoking TRPC1 activity, as has been shown with reference to magnetoreception.

## 10. ELF-EMF Parameters That Influence Cell Response

The importance of standardizing ELF-EMF parameters to be used for clinical exploitation has been previously underscored [68,172]. Fundamentally, ELF-EMF frequency, amplitude, and duration of exposure influence the developmental efficacy of exposure. Most studies have used time-variant magnetic fields in the amplitude range of 1–2 mT and typically in the frequency range of 30–60 Hz but many studies can be found using µT fields at radio-frequencies [5,16,17,18,19,21,22,23,30,33,44,58,59,61] (Figure 1). Another difference is whether the magnetic field is delivered at a continuous frequency or in a bursting pattern (as a barrage of spikes) [33]. Differences largely depend on the purpose of the exposure (proliferation versus apoptosis, etc.) as well as the magnetoreceptive model being tested. Exposure durations typically range from hours to days per week [5,16,17,18,19,21,22,23,30,33,44,59,61] with a few exceptions demonstrating developmental efficacy with brief exposures [10,65]. Nonetheless, it has been shown that static as well as time-variant magnetic fields are both capable of inducing mitohormetic responses based on ROS signaling [57,141], suggesting that diverse forms of magnetic exposure are recruiting similar transduction cascades. The symmetry of the magnetic signal around the 0 mT field, whether uniphasic (asymmetrical) or biphasic (symmetrical), and the rate of change (steepness; Tesla/second) and waveform of the applied signal (sine wave, square, triangular function or saw tooth) have also been shown to influence cellular response to magnetic exposure [10,49,59,61,62,65,66,67], but are seldom disclosed in published studies (Figure 2). Critically, the electrical signal applied to a coil system does not necessarily reflect the uniformity or magnetic wave form emanating from the coil system (Figure 2 and Figure 4), which, accordingly, will have important developmental consequences attributed to a particular coil system. Closer collaboration between qualified technical specialists and biologists is required in order to address and standardize these issues.

On the other hand, most published studies have not taken magnetic field orientation or directionality into consideration [58,173]. One way in which cells may perceive magnetic fields is via the magnitude of ionic currents induced by the magnetic fields. Predicted from physical principles, time-variant magnetic fields should induce ionic current flow at the membrane surface orthogonal to their plane of entry [174]. The greatest magnetic induction of ionic current will then be associated with the greatest degree of cross-alignment between the penetrating magnetic field lines and the major plane of cell spreading, independently of true vertical (up or down) or horizontal planes. In support of the ionic current induction model of magnetosensitivity, a contribution of cell alignment in magnetically induced developmental responses has been recently demonstrated in response to single, brief exposure [41].

The RPM also predicts an anisotropic interaction between an external static magnetic field with the electron spin moments of a flavoenzyme [146]. The observation that biological responses and, presumably, coherent electron spin dynamics, are modulated by the degree of cross-alignment between an imposed static magnetic field with an applied oscillating magnetic field serves as experimental validation of the RPM [40,142]. Therefore, the orientation of an applied oscillating magnetic field relative to both the long axis of cells/tissues and an external static magnetic field should be taken into consideration when optimizing for biological effect. Maximal magnetic responses would be then achieved when both the cell/tissue alignment and the magnetoresponsive moiety (for instance, a flavoenzyme) are situated in the appropriate orientation relative to the applied time-variant magnetic field. The question as to whether maximum responses are biologically beneficial, or detrimental, will ultimately depend on background biophysical stimulation in accordance with mitohormetic principles. Provocatively, the standing geomagnetic field may then serve as a reference frame with which to assign the direction of time-variant magnetic fields for optimal results. On the other hand, up- or down-directed fields should be perceived identically.

A differential effect of downward versus upward magnetic field exposure has been described with respect to non-uniform static magnetic fields (<1 mT; Figure 4) applied for a few days to cancer cells [37]. Notably, adherent cells responded differentially to upward- and downward-directed static magnetic fields, whereas cells in suspension did not respond irrespective of field directionality, suggesting that the substrate-mediated mechanotransduction was contributing to the background stimulation and assisted in achieving response threshold in response to static magnetic field exposure. It is important to note that in this case the cells were maintained in the presence of the aminoglycoside antibiotic streptomycin, which would reduce the sensitivity of cells to magnetic exposure (see previous, Section 5). A more recent study demonstrated that brief exposure (10 min) to uniform time-variant magnetic fields (1.5 mT) generated within a calibrated Helmholtz coil system (Figure 4) was capable of distinguishing the effects of magnetic field directionality in myoblasts in suspension and therefore independently of substrate-mediated shape-induced anisotropy [41]. In this second study, biological effects were more clearly resolved in response to downward field exposure, were recapitulated in adherent cells, and were, moreover, reproduced in several independent magnetic field devices. Notably, the corroborative data were obtained in the absence of the aminoglycoside antibiotics. This study also demonstrated that the effects of downward field exposure were stronger than those attributed to shape-induced anisotropy. The molecular mechanism through which downward field direction is distinguished from upward fields at the molecular level remains to be better understood.

Finally, induced electrical currents will vary with the rate of change of the applied time-variant magnetic field. Ionizing frequencies of EMFs (>10^16^ Hz; X-ray range, also see Figure 1) have been hypothesized to exert detrimental effects over cells and organisms due to enhanced oxidative stress largely arising from the radiolysis of water (superoxide anion (O_2_^•−^), hydroxyl radical (•OH), hydrated electron (e^−^_(aq)_), and hydrogen peroxide (H_2_O_2_)) [175]. Although X-ray radiation is outside the frequency range commonly used for EMF developmental studies, signal frequency would need to be taken into consideration when designing magnetic field protocols for best developmental outcomes.

## 11. Magnetic Nanoparticles and Nanocomposites

Magnetic nanoparticles, or nanocomposites, represent another magnetically tunable paradigm that would also benefit from calcium channel crosstalk. Magnetic nanoparticles may be employed to exert force on specifically targeted subcellular structures under the influence of an applied magnetic field [4,24,61,176]. Because of their paramagnetic nature, magnetic nanoparticles are inactive in the absence of magnetic field but can exert mechanical force on targeted subcellular structures in the direction of an applied magnetic field. Magnetic nanoparticles may be considered synthetic counterparts of biological magnetite (see Section 9). In one scenario, target specificity can be conferred by antibody binding to the nanoparticle following synthetic functionalization [176]. Magnetic nanoparticle utility would hence be evident in studies of mechanotransduction and, thus, effectively synergize with calcium channel-mediated mechanotransduction responses concomitantly recruited by magnetic exposure [4,24,61]. Other equally relevant applications of magnetic nanoparticles and nanocomposites have been adequately covered in existing reviews [4,24,61,176].

## 12. Conclusions and Future Perspectives

Magnetic fields are ever present in our environment and have been widely implicated in diverse developmental responses. Magnetism is just one of many forms of biophysical stimuli that may share certain aspects of a common transduction pathway by virtue of their recruitment of mitochondrial energy production to produce cellular adaptations via a process known as mitohormesis. These diverse forms of biophysical stimulation can combine their mitochondrial influences to produce a cumulative response that may confound interpretations of the generated data when based on the presumption of a single biophysical input. Commonly invoked by biophysical stimulation is a calcium–mitochondrial axis, wherein available evidence implicates a TRP channel class in combination with other calcium pathways. This calcium response effectively sets the mitohormetic backdrop on which cellular adaptations will be ultimately established. In this review we focused primarily on the participation of TRPC1 in magnetoreception based on its broad developmental and integrative nature as well as its reported close association with bioelectromagnetic phenomena. Therefore, in general, differences in cellular response to magnetic exposure arise from differences in mitochondrial involvement rather than actual differences in cellular machinery evoked by magnetic field exposure. Moreover, mitohormetic responses to magnetic stimulation will be influenced by differences in magnetic field parameters (amplitude, frequency, symmetry, duration of exposure, etc.), experimental environment (ambient magnetic fields, mechanical environment, temperature, pH, photons of light, aminoglycoside antibiotic usage, etc.), and cell developmental status (proliferative, differentiated, senescence, inflammatory status, etc.). Implicit to the experimental finding of “window effects”, field uniformity assumes the utmost importance in order to avoid “off-window” effects. Off-window effects have been shown in some studies to be counteractive to the sought-after effects associated with a particular field parameter. The broader implication is that the experimental environment will need to be taken into consideration when designing cellular or organismal therapeutic strategies, or, alternatively, extraneous biophysical and/or cellular parameters, as part of the experiment environment, would need to be adequately controlled for. As a “field”, we need to resist the pressure of having to come up with a “one size fits all” approach to electromagnetic therapy by adjusting our approaches to the experimental environment and by espousing the mitohormetic nature of electromagnetism. Finally, the presented model of magnetoreception is yet hypothetical, but consistent with available experimental findings, and is intended to provide a context for discussion and reflection, rather than represent a validated molecular arrangement.

## Figures and Tables

**Figure 1 bioengineering-10-01176-f001:**
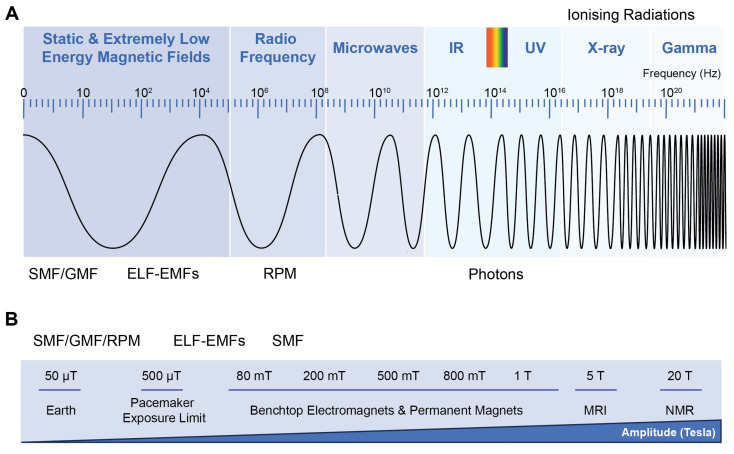
Most common frequencies and amplitudes of magnetic fields used in magnetoreceptive paradigms. (**A**) Magnetic field frequencies commonly employed in studies examining the biological effect of Static/Standing Magnetic Field (SMF), Geomagnetic Field (GMF), Extremely Low-Frequency Electromagnetic Fields (ELF-EMFs), and Radical Pair Mechanism (RPM), including photons of light. ELF-EMFs are also commonly referred to as Pulsed Electromagnetic Fields (PEMFs). Frequencies are shown against the backdrop of the entire electromagnetic spectrum. IR and UV are infrared and ultraviolet level radiations, respectively. Colors are the visible light range. (**B**) Range of magnetic field strengths commonly employed in studies examining the biological effect of SMF, GMF (~25–65 µT depending on location on the planet), ELF-EMFs, and RPM. This figure provides the most common values for magnetic field frequencies and amplitudes for the given applications; deviations from these values have been reported. Magnetic field strengths are given in Tesla (T) or Gauss (G). In this review, we employ Tesla for consistency with most published studies. 1 Tesla = 10^4^ Gauss. Panel B adapted from [4]. (Created with BioRender.com).

**Figure 2 bioengineering-10-01176-f002:**
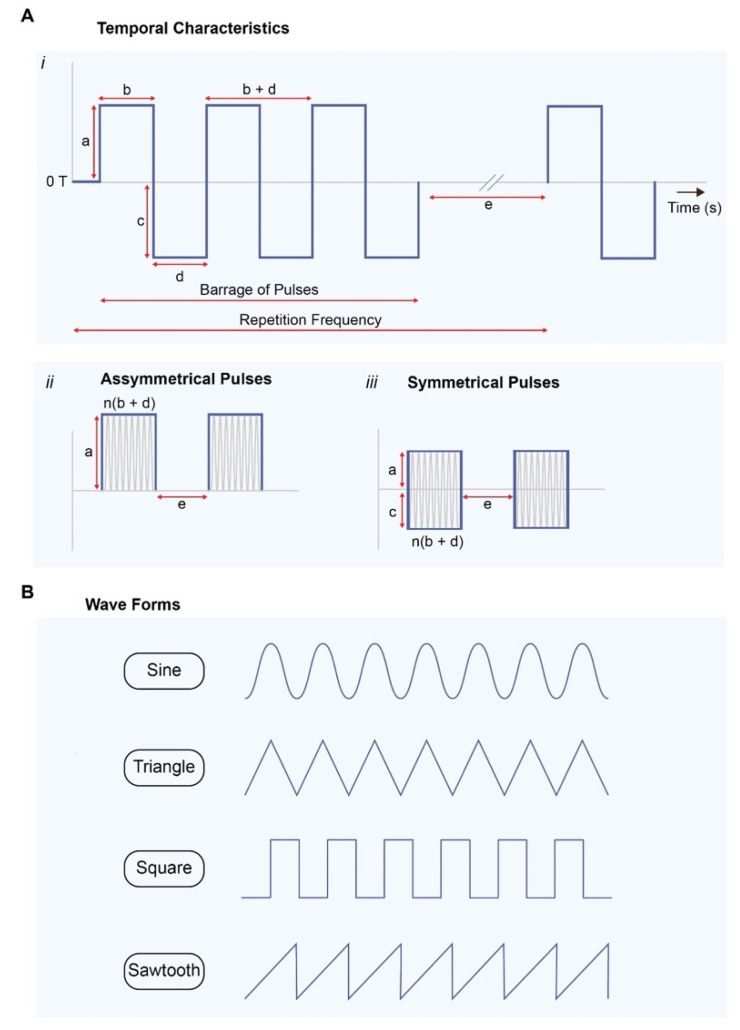
Distinct magnetic field symmetries and waveforms commonly employed in bioelectromagnetic studies. (**A*i***) Temporal characteristics of distinct PEMF signals where a = positive amplitude; b = positive width; c = negative amplitude; d = negative width. Shown is the electrical output from the signal generator, which, in ideal cases, approximates the magnetic field temporal characteristics, but is not always the case as it depends on accurate coil-electronic matching. (**A*ii***) Repetition frequency of a barrage of pulses. The signal becomes more continuous as e approaches d. In this example, the magnetic pulses are asymmetrical, starting and returning to zero magnetic field, without ever crossing zero magnetic field. (**A*ii***) Example of a symmetrical waveform. Both (**A*ii***) and (**A*iii***) have the same absolute magnitude (peak-to-trough), but (**A*iii***) has half the relative amplitude above and below 0 mT. Symmetrical signals also generate magnetic fields that change directions after crossing 0 mT, which would have developmental implications (see Section 10). (**B**) Distinct waveforms as indicated. A rapid rise time (Tesla/second) to the magnetic signal is a determinant feature for biological response [59] that will ultimately translate to the efficacy of a particular waveform. The different waveforms are derived from variations in this steepness factor. (Panel **A*i***) adapted from [68]. (Created with BioRender.com).

**Figure 3 bioengineering-10-01176-f003:**
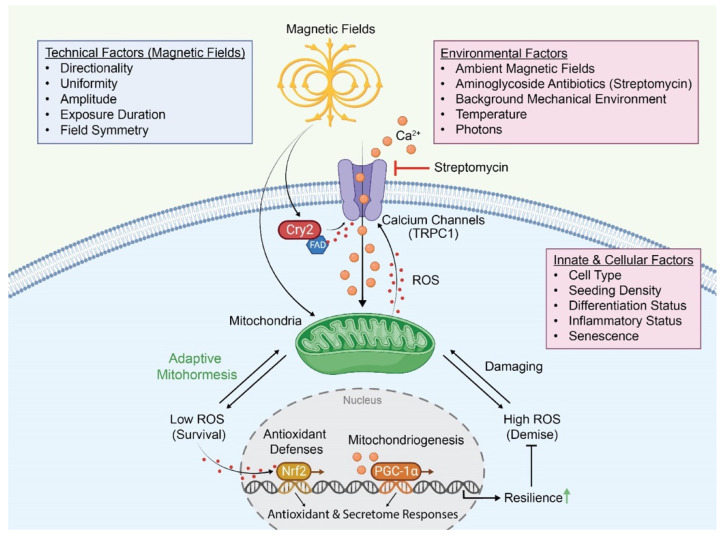
Hypothetical model of magnetoreception and the factors that may influence its operation. Orange circles represent Ca^2+^ ions and red circles represent ROS. Whether all aspects of the depicted enzymatic and transcriptional response cascade are recruited by all forms of magnetic stimulation, and in what tissues, remains to be scientifically determined. Environmental factors such as photons, temperature, and pH, as they influence enzymatic and mitochondrial efficiencies, as well as are detected by TRP channels, will influence magnetic mitohormetic responses. The transcriptional repercussions of magnetically stimulated Ca^2+^ and ROS increments over mitochondrial function were previously reviewed elsewhere [38]. (Created with BioRender.com).

**Figure 4 bioengineering-10-01176-f004:**
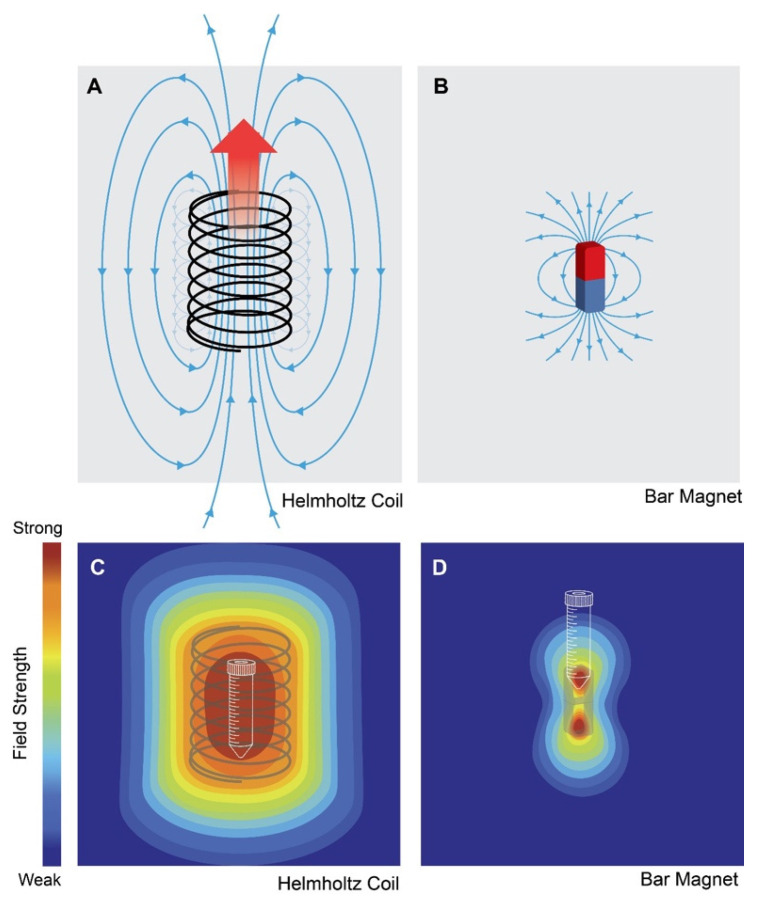
Differences in magnetic field uniformity obtained between a Helmholtz coil and a bar magnet. (**A**) Magnetic field lines generated with a Helmholtz coil. In this instance, clockwise electrical current flow through the coil produces an upwardly directed magnetic field. (**B**) Magnetic field lines associated with a bar magnet. Red and blue are north and south poles, respectively. (**C**) Field uniformity within a Helmholtz coil configuration. (**D**) Field uniformity associated with a bar magnet. A sample (conical tube) within the uniform region of a Helmholtz coil will receive uniform magnetic field exposure, whereas a sample at a distance from a point source of magnetic fields, such as a bar magnet, will experience non-uniform field exposure. (Created with BioRender.com).

## Data Availability

Not applicable.

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
