# Peer review of "Harmonizing Magnetic Mitohormetic Regenerative Strategies: Developmental Implications of a Calcium–Mitochondrial Axis Invoked by Magnetic Field Exposure"

_bioengineering, 2023, doi:10.3390/bioengineering10101176_

Round 1
Reviewer 1 Report
Here are my comments on the various points:
First of all, various acronyms are reported to describe different types of "electromagnetic fields" studied. For example, ”Extremely low-frequency electromagnetic fields (ELF-EMFs)”, “Pulsed Low-frequency EMF”, “Sinusoidal Low-frequency EMF”, Radio Frequency Magnetic Fields, Ambient-Supplemental Magnetic Fields, Static magnetic fields and static and RF magnetic fields at Zeeman resonance. This could create confusion in the non-expert readers, so the author must specify that these are not synonyms, but different type of EMF. For this reason, the different types of electromagnetic fields studied and used universally should be introduced in this review adding a new section in the introduction part.
The Authors take into consideration the effect of the geomagnetic field which strongly influences the electromagnetic exposure, but does not discuss it and does not report published studies which have been examined. This very important phenomenon has been extensively analyzed studied and reported in numerous articles (See Lisi at al. and Ledda at al.). Please add these new references in the article that report the proper control of the experiment environment, it also considers the role of temperature and lastly the effect on calcium channels during the exposure protocols.
It must also be revised for English fluency and typing errors.
Minor editing of English language required
Reviewer 2 Report
This review has some flasks, the authors must address before being published in this journal.
Authors ignored other stimuli factors such as pH, temperature, and light etc, and must add these factors
The authors should add experimental pictures, as they have discussed but there is no picture
Many grammatical errors throughout the manuscript
Authors add magnetic nanoparticle response to channels and drug delivery effect using magnetic field
nil
Reviewer 3 Report
This review focuses on delineating areas requiring technical and developmental harmonization to assist in the designing of therapeutic strategies and explores the underlying mechanistic similarities between magnetic field exposure and other forms of biophysical stimuli, such as mechanical stimuli, that mutually induce elevations in intracellular calcium and ROS as a prerequisite for biological outcome. As text descriptions, this review well summarizes featured facts. Such descriptions are valuable of being published in scientific journals including Bioengineering.
However, this manuscript has fatal problem. The numbers of figures and illustrations in this review are very less. Without sufficient guidance by visual items (figures and schemes). it is very tough for non-specialist to understand this review article. Therefore, I strongly request the authors to add at least five more figures. Without this change, I cannot recommend publication of this manuscript for publication in Bioengineering.
Round 2
Reviewer 2 Report
The authors addressed the questions
Reviewer 3 Report
It is OK. Acceptable.